# Glycated Albumin, a Novel Biomarker for Short-Term Functional Outcomes in Acute Ischemic Stroke

**DOI:** 10.3390/brainsci11030337

**Published:** 2021-03-06

**Authors:** Yerim Kim, Sang-Hwa Lee, Min Kyoung Kang, Tae Jung Kim, Han-Yeong Jeong, Eung-Joon Lee, Jeonghoon Bae, Kipyoung Jeon, Ki-Woong Nam, Byung-Woo Yoon

**Affiliations:** 1Department of Neurology, Kangdong Sacred Heart Hospital, Hallym University College of Medicine, Seoul 05355, Korea; 2Department of Neurology, Chuncheon Sacred Heart Hospital, Hallym University College of Medicine, Chuncheon 24253, Korea; neurolsh@hallym.or.kr; 3Department of Neurology, Uijeongbu Eulji Medical Center, Eulji University College of Medicine, Uijeonbu-si 11759, Korea; eiri616@hanmail.net; 4Department of Neurology, Seoul National University Hospital, Seoul National University College of Medicine, Seoul 03080, Korea; ttae35@gmail.com (T.J.K.); lejoon0824@gmail.com (E.-J.L.); baewjd@naver.com (J.B.); jkp0814@gmail.com (K.J.); 5Emergency Medical Center, Department of Neurology, Seoul National University Hospital, Seoul National University College of Medicine, Seoul 03080, Korea; hy123861@snu.ac.kr; 6Department of Neurology, Seoul Metropolitan Government-Seoul National University Boramae Medical Center, Seoul 03080, Korea; sdzero21@naver.com

**Keywords:** glycated albumin, glycated hemoglobin, brain ischemia, stroke, prognosis, biomarkers, blood glucose

## Abstract

*Background:* There is growing interest in the use of new biomarkers such as glycated albumin (GA), but data are limited in acute ischemic stroke. We explored the impact of GA on short-term functional outcomes as measured using the modified Rankin Scale (mRS) at 3 months compared to glycated hemoglobin (HbA1c). *Methods:* A total of 1163 AIS patients from two hospitals between 2016 and 2019 were included. Patients were divided into two groups according to GA levels (GA < 16% versus GA ≥ 16%). *Results:* A total of 518 patients (44.5%) were included in the GA ≥ 16% group. After adjusting for multiple covariates, the higher GA group (GA ≥ 16%) had a 1.4-fold risk of having unfavorable mRS (95% CI 1.02–1.847). However, HbA1c was not significantly associated with 3-month mRS. In addition, GA ≥ 16% was independently associated with unfavorable short-term outcomes only in patients without diabetes. *Conclusions:* In light of these results, GA level might be a novel prognostic biomarker compared to HbA1c for short-term stroke outcome. Although the impact of GA is undervalued in the current stroke guidelines, GA monitoring should be considered in addition to HbA1c monitoring.

## 1. Introduction

Glycated hemoglobin (HbA1c) reflects glycemic control over the past 2- to 3-month period and is considered the gold standard for the management of diabetes. Nonetheless, HbA1c has some clinical limitations; it does not reflect recent glycemic status, and a number of conditions (e.g., anemia, erythropoietin treatment, transfusion, and kidney disease) affect the test results [1,2]. Therefore, there is growing interest in using new biomarkers such as fructosamine or glycated albumin (GA), but there are limited data linking this indicator to outcomes in ischemic stroke. GA reflects glycemic control over approximately the last 2 to 4 weeks, reflecting the turnover of plasma proteins. Therefore, GA reflects relatively short-term control of glycemic status compared with HbA1c and could be a useful marker not influenced by situations that alter HbA1c levels. Additionally, because GA is measured by a standardized enzymatic methodology, the test is easy and fast to perform [3].

Multiple parameters, including initial glucose level, HbA1c, and glucose fluctuation, have been reported to predict poor stroke outcomes but remain controversial [3]. Traditionally, HbA1c is recognized as a well-known predictor of diabetic complications and cardiovascular outcomes [4,5,6,7]. However, several articles reported that HbA1c was not a good predictor of short-term outcomes, including in-hospital mortality, 6-month major adverse events, and all-cause mortality in the specific population [8,9]. In this regard, there has been some need to detect a novel index for diagnosing and managing glycemic control in covering the short-term effect in many critical illnesses. Furthermore, the role of GA in predicting stroke outcome in AIS has not been assessed previously. Therefore, we aimed to explore the impact of GA as a useful predictor for short-term functional outcome compared to HbA1c.

## 2. Materials and Methods

### 2.1. Study Population

We enrolled patients with acute ischemic stroke (AIS) or transient ischemic attack (TIA) who were admitted within 7 days of symptom onset to two stroke centers (Kangdong Sacred Heart Hospital, Hallym University College of Medicine, and Chuncheon Sacred Heart Hospital, Hallym University College of Medicine) from May 2016 through December 2019 into our prospective stroke registry system. Among a total of 1405 patients (541 patients from Kangdong Sacred Heart Hospital and 864 patients from Chuncheon Sacred Heart Hospital), fifty-eight subjects who were not evaluated for GA levels and 184 patients with TIA were excluded. As a result, a total of 1163 patients were included in this analysis (Figure 1).

All patients received standard and optimal medical therapy during hospitalization. The institutional review boards of the three centers (Kangdong Sacred Heart Hospital IRB no. 2020-02-006-001 and Chuncheon Sacred Heart Hospital IRB no. 2017-89) approved the study protocol, and written informed consent was obtained from all participants or from the next of kin when the patient’s agreement was not possible.

### 2.2. Clinical Information

All patients underwent diagnostic tests, including routine blood tests, neuroimaging, extracranial and intracranial vascular imaging, and cardiac studies. Demographic information including age and gender; stroke risk factors including hypertension (previous use of antihypertensive medication, systolic blood pressure > 140 mm Hg, or diastolic blood pressure >90 mm Hg at discharge), diabetes (previous use of antidiabetic medication under the diagnosis of diabetes, HbA1c ≥6.5%, fasting blood glucose >7.0 mmol/L (>126 mg/dL) at discharge), dyslipidemia (previous use of lipid-lowering medication, total cholesterol >6.0 mmol/L (>240 mg/dL), or low-density lipoprotein cholesterol >4.14 mmol/L (>160 mg/dL) at admission), smoking history, and atrial fibrillation; stroke subtypes according to the Trial of Org 10172 in Acute Stroke Treatment (TOAST) classification; and laboratory data. Body weight and height were measured upon admission. Body mass index (BMI) was calculated as the weight (kg) divided by the square of the height (m). Obesity status as a categorical variable was established by dividing BMI into four levels according to BMI quartiles (Q1, <21.57; Q2, 21.57–23.78; Q3, 23.78–25.97; and Q4, ≥25.97 kg/m^2^). The lowest BMI category was used as a reference level. GA levels were measured by an enzymatic method using albumin-specific proteinase and ketoamine oxidase (Beckman Coulter AU5821 Biochemical Analyzer; Tokyo, Japan). Since the reference interval for GA is different from what was observed in other studies [10,11,12], we divided the population into two groups (GA < 16% versus GA ≥ 16%).

Initial neurological severity (INS) was estimated using the National Institute of Health Stroke scale (NIHSS) score on admission. Based on the previous literature, a mild stroke was defined as NIHSS 0–7, while moderate to severe stroke was defined as NIHSS ≥ 8 [13]. The short-term functional outcome was estimated using the modified Rankin Scale (mRS) at 3 months after stroke onset. The short-term functional outcome was dichotomized (favorable outcome, 3-month mRS 0–2; unfavorable outcome, 3-month mRS 3–6).

### 2.3. Statistical Analysis

The distribution of demographic, clinical, laboratory, and stroke subtype data according to GA levels (GA < 16% versus GA ≥ 16%) was analyzed using the *χ*^2^ test and Student’s *t*-test, as appropriate. The trend in baseline data was also calculated using the *χ*^2^ test for trends in proportion. We also used one-way analysis of variance (ANOVA).

Values for the continuous variables are expressed as the means ± standard deviation (SD). Odds ratios (ORs) and 95% confidence intervals (CIs) were expressed for the results and probability values. A probability value of ≤0.05 was considered statistically significant. Analyses were performed using SPSS version 26.0 (SPSS Inc., Chicago, IL, USA).

### 2.4. Data Availability Statement

All data generated or analyzed during this study are included in this published article. Anonymized data will be shared by reasonable request from any qualified investigator.

## 3. Results

### 3.1. Analysis I: Levels of Glycated Albumin and Short-Term Functional Outcome

Among the 1163 subjects, the mean age was 69.7 ± 13.2 years, and 60.1% were men. The baseline demographic and clinical characteristics are shown in Table 1. A total of 518 patients (44.5%) were included in the GA ≥ 16% group. Patients with GA ≥ 16% were older and had some prevalent conventional vascular risk factors, such as prior ischemic stroke, hypertension, diabetes, and dyslipidemia. However, patients with GA ≥ 16% did not have high initial stroke severity, and unfavorable short-term functional outcomes (3-month mRS, 3–6) were prevalent (27.4% versus 39.7%) (Table 1).

After categorizing all patients into the two groups according to short-term functional outcome (favorable, 3-month mRS = 0–2, versus unfavorable, 3-month mRS = 3–6), patients with unfavorable short-term outcome were older, less obese, and more likely to have conventional vascular risk factors, including prior ischemic stroke, hypertension, and atrial fibrillation (Table 2). While HbA1c was not significantly different between the two groups (6.22 ± 1.33 versus 6.25 ± 1.37), GA levels were significantly elevated in subjects with unfavorable outcomes (16.30 ± 4.58 versus 17.20 ± 4.61). Figure 2A shows the distribution of short-term functional outcomes (3-month mRS) according to GA levels.

As the severity of obesity increased, the age decreased. As obesity severity increased, the level of HbA1c values, proportion of patients with DM, and proportion of patients with HbA1c ≥6.5% increased. However, the proportion of subjects with higher GA levels (*p* for trend = 0.003) and the GA/HbA1c ratio showed decreasing trends (Figure 2B and Appendix A).

We adjusted for age, sex, BMI, prior ischemic stroke history, hypertension, smoking, atrial fibrillation, anticoagulation treatment, TOAST classification, hemoglobin, triglyceride, blood urea nitrogen, high-sensitivity C-reactive protein (hsCRP), initial stroke severity, and GA (Table 3). After adjusting for multiple covariates, when compared to the lower GA group (GA < 16%), the higher GA group (GA ≥ 16%) had a 1.4-fold risk of having unfavorable short-term functional outcomes (OR 1.374; 95% CI 1.022–1.847). Increased age, prior ischemic stroke, TOAST classification, hsCRP, and initial stroke severity (NIHSS ≥8) were statistically significant predictors of unfavorable short-term outcomes.

In addition, analyses of the effects of several glycemic control parameters on short-term functional outcomes are shown in Table 4. When GA was replaced by other glycemic control parameters in the same model, only continuous levels of GA and GA ≥ 16% were significantly associated with unfavorable stroke outcomes at 3 months (3-month mRS 3–6), while continuous levels of HbA1c and HbA1c ≥ 6.5 failed to show a significant association.

In order to reduce the influence of people whose function was not good before the stroke, 44 subjects with pre-mRS scores of 4 or 5 were excluded. Among a total of 1119 patients, multivariate binary logistic regression was conducted. After adjusting for multiple covariates, when compared to the lower GA group (GA < 16%), the higher GA group (GA ≥ 16%) had a 1.4-fold risk of having unfavorable short-term functional outcomes (OR 1.399; 95% CI 1.033–1.894) (this table is not shown).

### 3.2. Analysis III: Levels of Glycoalbumin and Short-Term Functional Outcome by Glucose Tolerance Status

To investigate the relationship between GA and short-term functional outcome by glucose tolerance status, patients were classified into two groups based on diabetes history. In the binary logistic regression, after adjusting for multiple covariates, GA ≥ 16% was independently associated with unfavorable short-term outcomes only in patients without diabetes (Table 5).

## 4. Discussion

The main findings of this study were as follows: (1) patients with a higher GA level (GA ≥ 16%) had unfavorable short-term functional outcomes at 3 months after stroke onset; (2) a higher GA and higher GA/A1c ratio were significantly associated with unfavorable short-term functional outcomes at 3 months after stroke onset, but HbA1c was not; and (3) a higher GA level was associated with unfavorable short-term functional outcomes at 3 months after stroke onset only in patients without diabetes.

Several studies have demonstrated that GA is associated with vascular calcification and mortality [1,14]. In the Atherosclerosis Risk in Communities (ARIC) study of 11,104 participants, elevated baseline GA was significantly associated with cardiovascular outcomes even after adjustment for traditional risk factors, with especially strong associations in patients with diabetes mellitus (DM) [1]. In 49 hemodialysis subjects with type 2 DM, GA was significantly associated with the presence of peripheral vascular calcification and seems to be a better indicator of glycemic control than HbA1c [14]. However, few studies have evaluated the role of GA in IS patients [3,15]. In a subanalysis of the CHANCE (Clopidogrel in High-Risk Patients with Acute Nondisabling Cerebrovascular Events) trial, GA could be a potential marker to predict the effects of dual and single antiplatelet therapy on recurrent stroke [15]. In a total of 296 AIS patients with DM, higher GA (≥16%) was significantly associated with severe stroke (NIHSS > 14) and a large infarct volume [3]. Nevertheless, the relationship between GA and stroke outcome has not been assessed previously. In the present study, higher GA (≥16%) could increase the risk of unfavorable short-term functional outcomes after stroke. One explanation for this finding is that higher GA increases infarct volume [3]. According to previous reports, GA may be a good marker of macrovascular complications, while HbA1c is a good marker of microvascular complications [3]. In this regard, it is partially explained that large artery atherosclerosis (LAA) etiology accounts for a larger proportion in the group with higher GA in this study.

Interestingly, we found that when GA was replaced by HbA1c (either as a continuous variable or as a binary variable with a cutoff of HbA1c ≥ 6.5%) in the same model, HbA1c failed to prove the significant association in this study. We should not generalize this result. Conventionally, HbA1c has been a well-recognized predictor of cardiovascular outcomes in previous reports. Some authors demonstrated that HbA1c was a potential indicator for in-hospital death in patients with acute coronary syndrome [16] and was a good predictor of acute and long-term mortality in patients with AIS [17]. Furthermore, in a total of 534 subjects with AIS treated with mechanical thrombectomy, HbA1c ≥ 6.5% was an independent predictor of a poor outcome at 3 months after AIS [5]. However, the results are still controversial. Similar to our study, some prior articles reported that HbA1c was not an independent predictor of short-term outcomes [8,9]. In a retrospective study of 317 diabetic patients with acute coronary syndrome, HbA1c levels before admission were not related to short-term cardiovascular outcomes, including in-hospital mortality, 6-month major adverse cardiovascular events (MACE), and all-cause mortality [8]. In an observational multicenter study of 608 patients with acute myocardial infarction, HbA1c was not associated with 7-day mortality or 30-day mortality [9]. We cannot explain the exact pathomechanism, and we suggest that GA might also be an important indicator along with HbA1c [18].

Interestingly, we found that the effect of GA on short-term stroke outcome would differ according to concomitant DM. After classification by DM, the effect of GA on functional outcome at 3 months was significant only in patients without DM (OR 1.774, 95% CI 1.154–2.636). Consistent with our results, among all 2496 participants with AIS (2077 nondiabetic versus 419 diabetic), elevated admission glucose levels were associated with an increased risk for 30-day case fatality in patients without DM [19]. According to Huh et al., fasting glucose and postprandial glucose could influence the GA/A1c ratio in the prediabetes and type 2 DM groups, while those variables did not influence the GA/A1c ratio in the normal glucose tolerance group [20]. This result suggests that the influence of glycemic control parameters may be different according to glucose tolerance status. We hypothesized that in patients with previously identified DM, glucose-insulin homeostasis, drug interaction, and insulin resistance may also affect the result. This finding is meaningful because it suggests that since the DM state itself is an adverse factor, GA or GA/A1c cannot be an appropriate index without considering glycemic control status. Further well-designed studies to investigate the relationship between glycemic control indicators and glucose tolerance status are needed.

Interestingly, the effect of BMI on GA might also be considered to evaluate the role of GA on stroke outcome. According to previous reports, despite inconsistent results, obesity seems to be negatively associated with GA or GA/A1c [20,21,22,23]. In this study, BMI showed a significantly positive correlation with HbA1c, while BMI was negatively correlated with GA or GA/A1c (Appendix A). At present, the reasons for the negative influence of BMI on GA are not clear. However, a possible explanation is as follows: (1) Salas-Salvado et al. demonstrated that albumin levels are lower in obese subjects than in nonobese counterparts [24]. However, Miyashita et al. found that obese children had higher albumin concentrations than nonobese children [25], and Koga et al. reported no correlation between BMI and albumin levels [21]. (2) Turnover of albumin may be increased in obese patients. Since chronic low-grade systemic inflammation is involved in obese subjects, inflammation might increase the catabolic rate of albumin and decrease the rate of albumin synthesis. The authors did not provide the exact mechanism but hypothesized that inflammation represented by elevated hsCRP was significantly associated with BMI, which could result in lower GA [21]. In our study, when BMI increased, while GA showed a decreasing trend, albumin showed an increasing trend. Therefore, we suggest that it affects albumin turnover rather than the albumin concentration itself. Although we have not been able to draw any conclusions in this paper, the results suggest an important hypothesis for the next study.

The major strength is that this is a multicenter study with a relatively large sample size. In this regard, GA is expected to be a new glucose control marker in patients with AIS. However, several limitations should be noted in our study. First, this was a cross-sectional observational study design. Second, although we controlled for some confounders in this statistical model, some confounding factors may affect the result. Third, we did not evaluate all conditions affecting protein metabolism, such as thyroid dysfunction, liver cirrhosis, and nephrotic syndrome. Fourth, we could not present changes in blood sugar tests during admission or infarct growth at symptom aggravation. Therefore, we cannot provide direct evidence that GA, an indicator reflecting glucose fluctuation, is related to worse functional outcome. Finally, although we considered the effect of glucose tolerance status by DM status, we did not adjust for insulin resistance status because of the lack of serum insulin levels.

## 5. Conclusions

In conclusion, we suggest that GA levels are associated with short-term functional outcomes after AIS and might be a better prognostic biomarker than HbA1c. Although the role of GA in stroke outcome is undervalued in the current treatment guidelines, monitoring GA in addition to HbA1c could improve glycemic control in patients with AIS.

## Figures and Tables

**Figure 1 brainsci-11-00337-f001:**
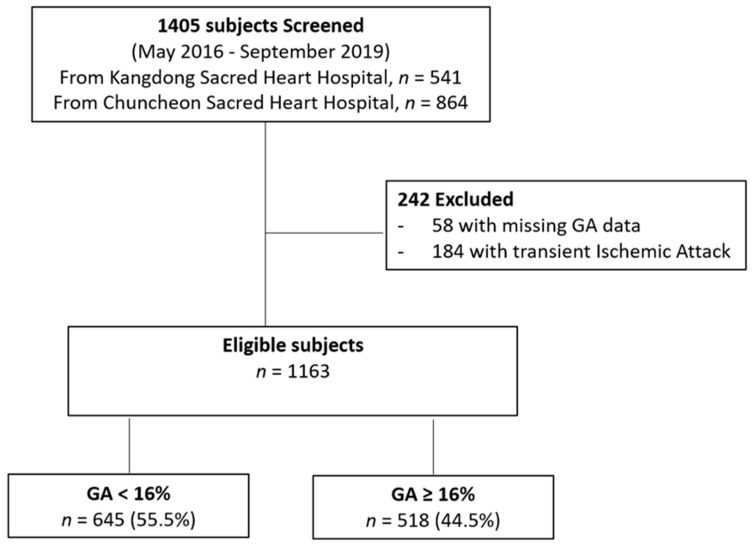
Flow diagram of study population. GA, glycated albumin.

**Figure 2 brainsci-11-00337-f002:**
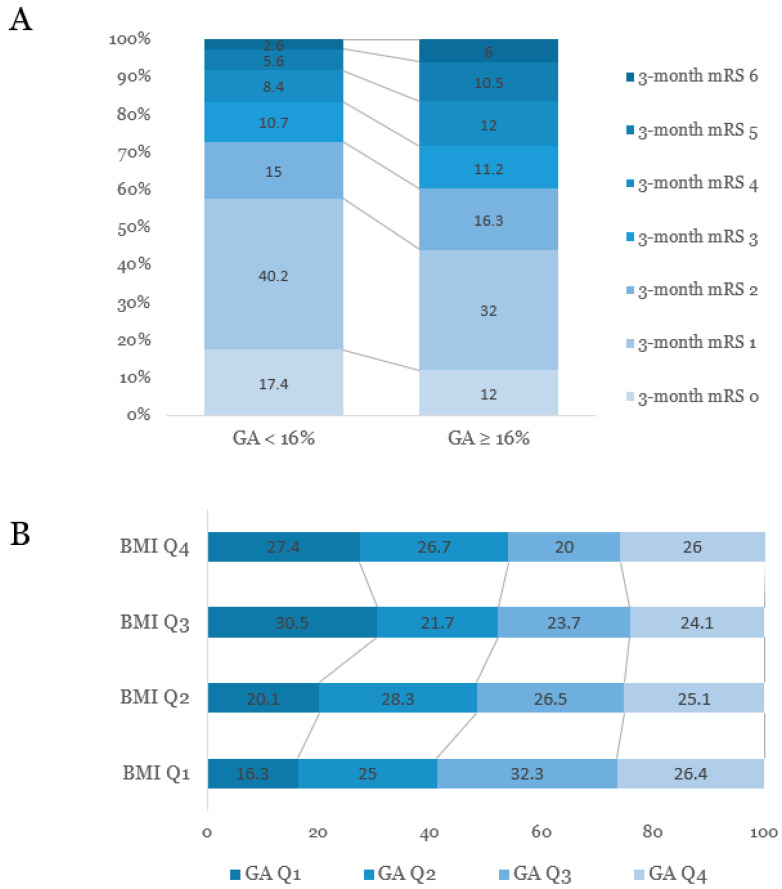
(**A**) Distribution of mRS scores at three months after stroke onset according to the levels of glycated albumin. (**B**) Associations between quartiles of glycated albumin and obesity status based on body mass index. GA, glycated albumin; BMI, body mass index; Q, quartile; mRS, modified Rankin Scale.

**Table 1 brainsci-11-00337-t001:** Baseline characteristics according to levels of glycated albumin.

	Glycated Albumin < 16%	Glycated Albumin ≥ 16%	*p*-Value
No. (%)	645 (55.5)	518 (44.5)	
Age, years	67.5 ± 13.5	72.4 ± 12.3	<0.001
Male sex, %	411 (63.7)	288 (55.6)	0.005
BMI at admission, kg/m^2^	23.94 ± 3.31	23.85 ± 3.82	0.670
Cardiovascular risk factor			
Prior ischemic stroke	109 (16.9)	137 (26.4)	<0.001
Hypertension	364 (56.4)	362 (69.9)	<0.001
Diabetes mellitus	92 (14.3)	318 (61.4)	<0.001
Dyslipidemia	81 (12.6)	88 (17.1)	0.031
Smoking	162 (25.1)	86 (16.6)	<0.001
Atrial fibrillation	99 (15.3)	101 (19.5)	0.078
Antiplatelet history	148 (22.9)	161 (31.1)	0.002
Anticoagulation history	28 (4.3)	31 (6.0)	0.056
Mechanism			0.001
LAA	202 (31.6)	209 (40.7)	
SVO	189 (29.5)	101 (19.6)	
Cardioembolic	107 (16.7)	90 (17.5)	
Other determined	32 (5.0)	19 (3.7)	
Undetermined	110 (17.2)	95 (18.5)	
Laboratory			
White Blood Cells	7769 ± 2678	7822 ± 2791	0.744
Hemoglobin, g/dL	13.9 ± 1.9	13.4 ± 2.1	<0.001
Platelet	233 K ± 68 K	224 K ± 72 K	0.028
FBS, mg/dL	117.4 ± 34.6	160.2 ± 73.5	<0.001
Initial glucose, mg/dL	124.6 ± 35.7	169.4 ± 74.1	<0.001
HbA1c, %	5.60 ± 0.60	7.00 ± 1.58	<0.001
HbA1c ≥ 6.5%	37 (5.8)	280 (54.4)	<0.001
Glycated albumin, %	13.8 ± 1.3	20.1 ± 4.8	<0.001
GA/HbA1c	2.47 ± 0.33	2.88 ± 0.48	<0.001
Initial glucose/GA	8.96 ± 2.62	8.28 ± 2.98	<0.001
Initial glucose/HbA1c	22.14 ± 6.46	23.63 ± 7.88	0.001
LDL, mg/dL	102.0 ± 32.9	100.7 ± 37.2	0.530
Total cholesterol, mg/dL	170.1 ± 42.6	164.0 ± 45.2	0.019
Triglyceride, mg/dL	137.1 ± 105.1	135.1 ± 81.4	0.729
Blood urea nitrate	16.97 ± 11.05	19.31 ± 9.93	<0.001
Creatinine	0.94 ± 0.65	1.07 ± 0.89	0.009
Albumin	3.98 ± 0.43	3.93 ± 0.47	0.072
Prothrombin Time	1.08 ± 0.50	1.08 ± 0.22	0.113
hsCRP	7.90 ± 24.00	14.14 ± 34.32	0.001
Systolic BP, mmHg	151 ± 28	148 ± 28	0.050
Diastolic BP, mmHg	86 ± 15	84 ± 16	0.023
Initial Stroke Severity, NIHSS 0–7	478 (74.1)	404 (78.0)	0.024
NIHSS ≥ 8	167 (25.9)	114 (22.0)	
Poor short-term functional outcome(3-month mRS, 3–6)	166 (27.4)	192 (39.7)	<0.001

*Note:* Abbreviations: BMI, body mass index; FBS, fasting blood sugar; GA, glycated albumin; LDL, low-density lipoprotein; HDL, high-density lipoprotein; aPTT, activated prothrombin time; LAA, large artery atherosclerosis; SVO, small vessel occlusion; hsCRP, high-sensitivity C-reactive protein; BP, blood pressure; IQR, interquartile range; NIHSS, National Institutes of Health Stroke Scale. No. (%) or mean ± SD. *p*-values were calculated by the χ^2^ test for trend in proportions.

**Table 2 brainsci-11-00337-t002:** Baseline characteristics according to short-term functional outcomes.

	Good Functional Outcome 3-Month mRS, 0–2	Poor Functional Outcome3-Month mRS, 3–6	*p*-Value
No. (%)	731 (67.1)	358 (32.9)	
Age, years	67 ± 13	75 ± 12	<0.001
Male sex, %	475 (65.0)	176 (49.2)	<0.001
BMI at admission, kg/m^2^	24.09 ± 3.38	23.53 ± 3.88	0.015
BMI Quartile, Q1	166 (23.0)	101 (28.6)	0.015 *
Q2	170 (23.6)	91 (25.8)	
Q3	195 (27.0)	83 (23.5)	
Q4	190 (26.4)	78 (22.1)	
Cardiovascular risk factor			
Prior ischemic stroke	128 (17.5)	101 (28.2)	<0.001
Hypertension	429 (58.7)	240 (67.0)	0.008
Diabetes mellitus	244 (33.4)	138 (38.5)	0.093
Dyslipidemia	89 (12.2)	59 (16.5)	0.051
Smoking	176 (24.1)	47 (13.1)	<0.001
Atrial fibrillation	87 (11.9)	94 (26.3)	<0.001
Antiplatelet history	178 (24.4)	111 (31.0)	0.055
Anticoagulation history	18 (2.5)	33 (9.2)	<0.001
Mechanism			<0.001
LAA	250 (34.5)	135 (37.8)	
SVO	223 (30.8)	57 (16.0)	
Cardioembolic	100 (13.8)	82 (23.0)	
Other determined	32 (4.4)	15 (4.2)	
Undetermined	119 (16.4)	68 (19.0)	
Laboratory			
White Blood Cells	7676 ± 2542	8019 ± 3117	0.071
Hemoglobin, g/dL	13.9 ± 1.8	13.2 ± 2.2	<0.001
Platelet	230K ± 64K	228K ± 80K	0.570
FBS, mg/dL	134.4 ± 59.6	139.6 ± 59.1	0.174
Initial glucose, mg/dL	143.8 ± 61.1	145.6 ± 59.9	0.656
HbA1c, %	6.22 ± 1.33	6.25 ± 1.37	0.678
HbA1c ≥6.5%	202 (27.7)	102 (28.7)	0.715
Glycated albumin, %	16.30 ± 4.58	17.20 ± 4.61	0.002
Glycated albumin ≥ 16%	292 (39.9)	192 (53.6)	<0.001
GA/HbA1c	2.62 ± 0.40	2.74 ± 0.51	<0.001
Initial glucose/GA	8.85 ± 2.47	8.56 ± 2.77	0.073
Initial glucose/HbA1c	22.89 ± 6.42	23.26 ± 7.08	0.383
LDL, mg/dL	102.5 ± 35.1	99.9 ± 35.5	0.257
Total cholesterol, mg/dL	168.8 ± 43.0	164.3 ± 45.5	0.118
Triglyceride, mg/dL	144.7 ± 96.7	121.8 ± 96.5	<0.001
Blood urea nitrate	17.15 ± 9.52	19.68 ± 11.18	<0.001
Creatinine	0.98 ± 0.73	1.07 ± 0.88	0.129
Albumin	4.00 ± 0.40	3.85 ± 0.52	<0.001
Prothrombin Time	1.08 ± 0.49	1.09 ± 0.19	0.714
hsCRP	7.63 ± 23.63	17.10 ± 38.31	<0.001
Systolic BP, mmHg	149 ± 27	150 ± 29	0.686
Diastolic BP, mmHg	85 ± 15	84 ± 17	0.095
Initial Stroke Severity, NIHSS 0–7	592 (81.0)	237 (66.2)	<0.001
NIHSS ≥ 8	139 (19.0)	121 (33.8)	

*Note:* Abbreviations: BMI, body mass index; FBS, fasting blood sugar; GA, glycated albumin; LDL, low-density lipoprotein; HDL, high-density lipoprotein; aPTT, activated prothrombin time; LAA, large artery atherosclerosis; SVO, small vessel occlusion; hsCRP, high-sensitivity C-reactive protein; BP, blood pressure; IQR, interquartile ratio; NIHSS, National Institutes of Health Stroke Scale. No. (%) or mean ± SD. *p*-values were calculated by the χ^2^ test for trend in proportions. * Linear-by-linear association for trend was used.

**Table 3 brainsci-11-00337-t003:** Effect of GA on unfavorable short-term outcomes (compared to favorable 3-month mRS 0–2).

Variables	OR	95% CI	*p*-Value
Age, per 1 year	1.034	1.019–1.050	<0.001
Male sex	0.728	0.525–1.010	0.057
BMI at admission, kg/m^2^	1.000	0.958–1.043	0.994
Cardiovascular risk factor			
Prior ischemic stroke	1.610	1.137–2.278	0.007
Hypertension	0.902	0.652–1.248	0.533
Smoking	0.930	0.610–1.418	0.736
Atrial fibrillation	1.528	0.826–2.826	0.177
Anticoagulation treatment	0.691	0.038–12.562	0.803
Mechanism			
LAA	1.674	1.128–2.484	0.011
SVO	Reference		
Cardioembolic	1.215	0.612–2.410	0.578
Other determined	1.732	0.766–3.921	0.187
Undetermined	1.454	0.904–2.340	0.123
Laboratory			
Hemoglobin	0.988	0.906–1.078	0.792
Triglyceride	0.999	0.998–1.001	0.447
Blood urea nitrogen	1.008	0.993–1.022	0.294
Albumin	0.854	0.589–1.239	0.405
hsCRP	1.007	1.001–1.012	0.012
Glycated albumin <16%	Reference		
Glycated albumin ≥ 16%	1.374	1.022–1.847	0.035
Initial Stroke Severity, NIHSS 0–7	Reference		
NIHSS ≥ 8	1.875	1.345–2.613	<0.001

*Note:* Adjusted for age, sex, body mass index, prior ischemic stroke, hypertension, smoking, atrial fibrillation, anticoagulation treatment, stroke subtype, hemoglobin, triglyceride, blood urea nitrogen, albumin, hsCRP, initial stroke severity, and glycated albumin. Abbreviations: BMI, body mass index; hsCRP, high-sensitivity C-reactive protein; mRS, modified Rankin Scale; OR, odds ratio; CI, confidence interval; NIHSS, National Institutes of Health Stroke Scale.

**Table 4 brainsci-11-00337-t004:** Effect of glycemic control parameters on unfavorable short-term outcomes (compared to favorable 3-month mRS 0–2).

Variables	OR	95% CI	*p*-Value
HbA1c	1.078	0.967–1.203	0.176
HbA1c ≥ 6.5%	1.233	0.889–1.711	0.210
GA/HbA1c	1.395	0.990–1.966	0.057
Glycated albumin	1.037	1.005–1.069	0.022
Glycated albumin ≥ 16%	1.374	1.022–1.847	0.035

*Note:* Adjusted for age, sex, hypertension, diabetes, dyslipidemia, smoking, atrial fibrillation, stroke subtype, glycated albumin, and body mass index. Abbreviations: GA, glycated albumin; OR, odds ratio; CI, confidence interval.

**Table 5 brainsci-11-00337-t005:** Effect of GA on unfavorable short-term outcomes (compared to favorable 3-month mRS 0-2) based on the presence of diabetes.

	Patients with Diabetes			Patients without Diabetes		
Variables	OR	95% CI	*p*-Value	OR	95% CI	*p*-Value
Age, per 1 years	1.036	1.009–1.064	0.009	1.034	1.015–1.054	<0.001
Male sex	0.717	0.417–1.233	0.229	0.705	0.461–1.078	0.106
BMI at admission, kg/m^2^	1.004	0.930–1.083	0.917	0.996	0.944–1.051	0.890
Cardiovascular risk factor						
Prior ischemic stroke	1.690	0.977–2.924	0.060	1.565	0.983–2.491	0.059
Hypertension	0.970	0.526–1.791	0.923	0.893	0.599–1.329	0.576
Smoking	0.946	0.462–1.936	0.880	0.927	0.539–1.596	0.785
Atrial fibrillation	0.354	0.096–1.299	0.117	2.419	1.108–5.282	0.027
Anticoagulation treatment	0.900	0.274–2.956	0.862	3.050	1.061–8.771	0.039
Mechanism						
LAA	2.653	1.349–5.218	0.005	1.321	0.802–2.176	0.275
SVO	Reference			Reference		
Cardioembolic	10.593	2.353–47.683	0.002	0.599	0.254–1.408	0.240
Other determined	2.634	0.425–16.321	0.298	1.518	0.587–3.924	0.389
Undetermined	2.913	1.278–6.639	0.011	0.984	0.538–1.799	0.958
Laboratory						
Hemoglobin	0.962	0.835–1.108	0.588	1.041	0.923–1.174	0.517
Triglyceride	0.999	0.996–1.002	0.438	1.000	0.997–1.002	0.660
Blood urea nitrogen	1.009	0.984–1.035	0.476	1.004	0.986–1.023	0.660
Albumin	0.758	0.417–1.378	0.364	0.852	0.516–1.406	0.530
hsCRP	1.006	0.997–1.016	0.199	1.007	1.000–1.013	0.039
Glycoalbumin < 16%	Reference			Reference		
Glycoalbumin ≥ 16%	0.820	0.456–1.472	0.506	1.744	1.154–2.636	0.008
Initial Stroke Severity, NIHSS 0–7	Reference			Reference		
NIHSS ≥ 8	1.461	0.807–2.646	0.211	2.082	1.373–3.156	0.001

*Note:* Adjusted for age, sex, hypertension, diabetes, dyslipidemia, smoking, atrial fibrillation, stroke subtype, glycated albumin, and body mass index. Abbreviations: BMI, body mass index; LAA, large artery atherosclerosis; SVO, small vessel occlusion; mRS, modified Rankin Scale; OR, odds ratio; CI, confidence interval.

## Data Availability

All data generated or analyzed during this study are included in this published article. Anonymized data will be shared by reasonable request from any qualified investigator.

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
