# Peer review of "Glycated Albumin, a Novel Biomarker for Short-Term Functional Outcomes in Acute Ischemic Stroke"

_brainsci, 2021, doi:10.3390/brainsci11030337_

Round 1

Reviewer 1 Report

Major Comments:

Introduction: It is unclear why the authors chose to incorporate the inverse relation of GA with the BMI in the title and its relation to this paper which pertains to GA and stroke outcomes.

Of note, the authors do not discuss or even mention BMI in the introduction. There has been decent literature already showing the inverse relation of GA with BMI (below) and this information seems irrelevant in this manuscript setting.

He X, Mo Y, Ma X, Ying L, Zhu W, Wang Y, Bao Y, Zhou J. Associations of body mass index with glycated albumin and glycated albumin/glycated hemoglobin A1c ratio in Chinese diabetic and non-diabetic populations. Clin Chim Acta. 2018 Sep;484:117-121. doi: 10.1016/j.cca.2018.05.044. Epub 2018 May 24. PMID: 29802831.

Miyashita Y, Nishimura R, Morimoto A, Matsudaira T, Sano H, et al. (2007) Glycated albumin is low in obese, type 2 diabetic patients. Diabetes Res Clin Pract 78: 51–55.

Nishimura R, Kanda A, Sano H, Matsudaira T, Miyashita Y, et al. (2006) Glycated albumin is low in obese, non-diabetic children. Diabetes Res Clin Pract 71: 334–338.

Koga M, Matsumoto S, Saito H, Kasayama S (2006) Body mass index negatively influences glycated albumin, but not glycated hemoglobin, in diabetic patients. Endocr J 53: 387–391.

HbA1c has been consistently shown to have an association with poor outcomes in AIS (several references below). Interestingly the articles that the authors reference with their statement pertaining to A1c not being a good predictor of outcomes are related to non-stroke pathologies (references 4 and 5) and one study they did reference with the A1c (reference 3) actually showed that high HbA1c was associated with increasing infarct volume.

HbA1c (Glycated Hemoglobin) Levels and Clinical Outcome Post-Mechanical Thrombectomy in Patients With Large Vessel Occlusion

Kang-Ho Choi, MD, PhD, Ja-Hae Kim, MD, PhD, Kyung-Wook Kang, MD, Joon-Tae Kim, MD, PhD, Seong-Min Choi, MD, PhD, Seung-Han Lee, MD, PhD, Man-Seok Park, MD, PhD, Byeong-Chae Kim, MD, PhD, Myeong-Kyu Kim, MD, PhD, Ki-Hyun Cho, MD, PhD Stroke 2019

Wang, Hong et al. Impact of Elevated Hemoglobin A1c Levels on Functional Outcome in Patients with Acute Ischemic Stroke. Journal of Stroke and Cerebrovascular Diseases, Volume 28, Issue 2, 470 - 476

Glycosylated Hemoglobin A1c Predicts Intracerebral Hemorrhage with Acute Ischemic Stroke Post-Mechanical Thrombectomy  Sun, Chenghe et al.

Journal of Stroke and Cerebrovascular Diseases, Volume 29, Issue 9, 105008

Diprose WK, Wang MTM, McFetridge A, et al

Glycated hemoglobin (HbA1c) and outcome following endovascular thrombectomy for ischemic stroke

Journal of NeuroInterventional Surgery 2020;12:30-32.

Clara Hjalmarsson, Karin Manhem, Lena Bokemark, Björn Andersson, "The Role of Prestroke Glycemic Control on Severity and Outcome of Acute Ischemic Stroke", Stroke Research and Treatment, vol. 2014, Article ID 694569, 6 pages, 2014. https://doi.org/10.1155/2014/694569

Methods:

The inclusion of TIAs is problematic when comparing the MRS at 3 months since technically this should not contribute to a change in the MRS. Of note, the authors report the baseline severity percentages (non-significant difference), I wonder if the stratification of TIAs would have made a difference since if a major proportion of patients in the < 16 GA group were low NIH of 0-2 and the ones in the high GA group were in the relatively worse baseline like NIH of 5-7 this could have contributed to worse outcomes as well. This stratification applies to NIH of >8 as well considering worse glucose control in days preceding stroke presentation has been associated with worse Ischemic stroke presentations.

Results:

  1. Whereas patients with GA≥16% did not have severe initial stroke severity, and shot-term unfavorable functional outcome (3 month mRS, 3-6) was prevalent (25.7% versus 38.6%) (Table 1).

There was no significant difference in the univariate analysis and so this interpretation of not having severe stroke severity cannot be done. Please see above comment in the methods section.

  1. b) After categorizing all patients into the two groups according to short-term functional outcome (favorable, 3-month mRS= 0-2, versus unfavorable, 3-month mRS= 3-6), patients with unfavorable short-term outcome were older, less obese, and more likely to have conventional vascular risk factors including prior ischemic stroke, hypertension, diabetes, and atrial fibrillation (Table 2).

In the final multivariate regression analysis, (Table 3) the authors left out diabetes (which was subgrouped later), use of antiplatelets, anticoagulants, all of which could potentially have influenced the outcomes from a stroke standpoint and were significantly higher in the GA> 16 group based on Table 2. This could pose a challenge in the interpretation of results.

Table 3: Can the authors explain how these specific variables were chosen?

The information on GA and BMI as stated prior is irrelevant in this manuscript:

He X, Mo Y, Ma X, Ying L, Zhu W, Wang Y, Bao Y, Zhou J. Associations of body mass index with glycated albumin and glycated albumin/glycated hemoglobin A1c ratio in Chinese diabetic and non-diabetic populations. Clin Chim Acta. 2018 Sep;484:117-121. doi: 10.1016/j.cca.2018.05.044. Epub 2018 May 24. PMID: 29802831.

Miyashita Y, Nishimura R, Morimoto A, Matsudaira T, Sano H, et al. (2007) Glycated albumin is low in obese, type 2 diabetic patients. Diabetes Res Clin Pract 78: 51–55.

Nishimura R, Kanda A, Sano H, Matsudaira T, Miyashita Y, et al. (2006) Glycated albumin is low in obese, non-diabetic children. Diabetes Res Clin Pract 71: 334–338.

Koga M, Matsumoto S, Saito H, Kasayama S (2006) Body mass index negatively influences glycated albumin, but not glycated hemoglobin, in diabetic patients. Endocr J 53: 387–391.

Discussion:

  1. 1) patients with a higher GA level 202 (GA≥16%) had unfavorable short-term functional outcomes at 3 month after stroke onset; 203 (2) a higher GA and higher GA/A1c ratio was significantly associated with unfavorable 204 short-term functional outcome at 3 month after stroke onset, but HbA1c was not; and (3) a higher GA level was associated with unfavorable short-term functional outcome at 3 months after stroke onset only in patients without diabetes.

Based on the aforementioned comments in previous sections, the inference 1 cannot be sustained, inference 3 is a result of subgroup testing without multiple correction. The authors can consider the use of Bonferroni correction or similar strategies to overcome this issue.

  1. Similar to our study, some previous articles reported that HbA1c was not an independent predictor of short-term outcomes [4,5]. In a retrospective study of 317 diabetic patients with acute coronary syndrome, HbA1c levels before admission are not related to short-term cardiovascular outcome including in-hospital mortality, 6-month major adverse cardiovascular events (MACE), and all-cause mortality [4]. In an observational multicenter study of 608 patients with acute myocardial infarction, HbA1c was not associated with 7 day mortality and 30-day mortality [5]. However, the results are still controversial. Some authors demonstrated that HbA1c was a potential indicator for in-hospital death in patients with acute coronary syndrome [12] and was a good predictor of acute and long-term mortality in patients with AIS [13]. Furthermore, in a total of 534 subjects with AIS treated with mechanical thrombectomy, HbA1c≥6.5% was an independent predictor of a poor outcome at 3 months after AIS [14].

As stated prior including references this is not accurate from a stroke standpoint. Can the authors respond why cardiovascular outcomes were chosen instead of stroke outcomes when there is good data on stroke-based outcomes and A1c relation? Please see information from the introduction and the references associated with stroke outcomes and A1c. Of note, the one reference from an AIS standpoint actually showed A1cs relation with poor outcome. Reference 14. This is one of the fewer studies to show A1c to not be related to poor outcomes, but the interpretation is limited due to the aforementioned limitations and potential confounding.

  1. In addition, the effect of BMI on GA might also be considered to evaluate the role of 259 GA on stroke outcome. According to previous reports, in spite of the inconsistent results, 260 obesity seems to be negatively associated with GA or GA/A1c [17-20]. In this study, the 261 BMI showed a significantly positive correlation with HbA1c, while BMI negatively corre-262 lated with GA or GA/A1c (supplementary table). At present, the reasons for the negative 263 influence of BMI on GA are not clear. However, possible explanation is as follows: (1) 264 Salas-Salvado et al. demonstrated that albumin levels are lower in obese subjects than 265 non-obese counterparts [21]. But Miyashita et al. found that obese children had higher 266 albumin concentration than non-obese children [22], and Koga et al. reported no correla-267 tion between BMI and albumin levels [18]. (2) turnover of albumin may be increased in 268 obese patients. Since chronic low-grade systemic inflammation is involved in obese sub-269 jects, inflammation might increase the catabolic rate of albumin and decrease the rate of 270 albumin synthesis. The authors did not provide the exact mechanism, but hypothesized 271 that inflammation represented by elevated hs-CRP was significantly associated with BMI, 272 which could result in lower GA [18]. In our study, when BMI is increasing, while GA 273 showed a decreasing trend, albumin showed an increasing trend. Therefore, we suggest 274 that it affects albumin turnover rather than the albumin concentration itself. Although we 275 have not been able to draw any conclusions through this paper, I think it suggested an 276 important key finding for the next study.

As stated, this discussion of BMI and GA relation does not seem relevant to their primary hypothesis and goal of this paper.

  1. A major strength is that this is the first study to evaluate the effect of GA on short- term functional outcome in patients with AIS.

The below study was published on GA and outcomes in AIS late last year and so this sentence can bebe tempered.

Lee SH, Kim Y, Park SY, Kim C, Kim YJ, Sohn JH. Pre-Stroke Glycemic Variability Estimated by Glycated Albumin Is Associated with Early Neurological Deterioration and Poor Functional Outcome in Prediabetic Patients with Acute Ischemic Stroke. Cerebrovasc Dis. 2021;50(1):26-33. doi: 10.1159/000511938. Epub 2020 Dec 1. PMID: 33260185.

Grammatical / editing / Minor Comments

“growing interest to use a new biomarker such as Glycated albumin (GA), but data are limited in acute ischemic stroke”

“We explored the impact of GA for short-term 18 functional outcome using the modified Rankin Scale (mRS) at 3 months compared to glycated hemoglobin (HbA1c);”

“The 595 patients (44.2%) were included in the GA≥16% group.”

“GA level might be a novel prognostic biomarker than HbA1c for short-term stroke outcome. Although the impact of GA is undervalued in the current stroke guidelines, monitoring GA should be considered in addition to HbA1c;”

“In this regard, there have been some needs for detecting a novel index for diagnosing and managing glycemic control in covering the short-term effect in many critical illnesses.”

“fifty-eight subjects who were not evaluate for GA levels were excluded”

Further well-designed study to investigate the relationship between glycemic control indicators and glucose tolerance status should be needed.

Patients with GA≥16% were older and had some prevalent conventional vascular risk factors, such as prior ischemic stroke, hypertension, diabetes, smoking history, and atrial fibrillation. Whereas patients with GA≥16% did not have severe initial stroke severity, and shot-term unfavorable functional outcome (3 month mRS, 3-6) was prevalent (25.7% versus 38.6%)

Overall:

Multiple issues with Grammar - article usage, punctuations and sentence structuring through the manuscript.  Only a few examples have been elicited above. Before re-submitting this manuscript, the authors could consider having it proofread by someone well-versed in written English or possibly utilize an editing resource. Brevity is also an issue with this manuscript, with an extensive amount of irrelevant information including some on BMI and GA which doesn’t pertain to the primary aim of this manuscript. Some of the biggest limitations pertain to the methodological aspects and the confounding as discussed above and the inaccurate discussion regarding A1c and outcomes.

Author Response

Journal : Brain Sciences (ISSN 2076-3425)

Manuscript ID : brainsci-1108388

Type : Article

Title : Glycated Albumin, a Novel Biomarker for Short-Term Stroke Outcome and Its Inversed Association with Body Mass Index

We thank the reviewer for the valuable feedback, which enabled us to greatly improve the quality of our manuscript. Here we address the reviewer’ concerns point-by-point and explain, where applicable, how we have been able to incorporate the suggestions into our revision. The reviewer’s original comments are copied below in italics.

Major Comments:

  1. Introduction: It is unclear why the authors chose to incorporate the inverse relation of GA with the BMI in the title and its relation to this paper which pertains to GA and stroke outcomes.

Of note, the authors do not discuss or even mention BMI in the introduction. There has been decent literature already showing the inverse relation of GA with BMI (below) and this information seems irrelevant in this manuscript setting.

He X, Mo Y, Ma X, Ying L, Zhu W, Wang Y, Bao Y, Zhou J. Associations of body mass index with glycated albumin and glycated albumin/glycated hemoglobin A1c ratio in Chinese diabetic and non-diabetic populations. Clin Chim Acta. 2018 Sep;484:117-121. doi: 10.1016/j.cca.2018.05.044. Epub 2018 May 24. PMID: 29802831.

Miyashita Y, Nishimura R, Morimoto A, Matsudaira T, Sano H, et al. (2007) Glycated albumin is low in obese, type 2 diabetic patients. Diabetes Res Clin Pract 78: 51–55.

Nishimura R, Kanda A, Sano H, Matsudaira T, Miyashita Y, et al. (2006) Glycated albumin is low in obese, non-diabetic children. Diabetes Res Clin Pract 71: 334–338.

Koga M, Matsumoto S, Saito H, Kasayama S (2006) Body mass index negatively influences glycated albumin, but not glycated hemoglobin, in diabetic patients. Endocr J 53: 387–391.

Response: Thank you for your valuable comments.

I know there were already several articles showing the inverse relationship between GA and BMI.

However, few articles have dealt with the relationship between BMI and GA in patients with Acute ischemic stroke.

The authors thought this was very interesting and added it to the title.

However, as the reviewer points out, the authors fully agree that this title does not serve the purpose of this paper. Thank you for raising so much important comments.

So we changed the title as follows.

Glycated Albumin, a Novel Biomarker for Short-Term Stroke Outcome and Its Inversed Association with Body Mass Index

àGlycated Albumin, a Novel Biomarker for Short-Term Functional Outcomes In Acute Ischemic Stroke

  1. HbA1c has been consistently shown to have an association with poor outcomes in AIS (several references below). Interestingly the articles that the authors reference with their statement pertaining to A1c not being a good predictor of outcomes are related to non-stroke pathologies (references 4 and 5) and one study they did reference with the A1c (reference 3) actually showed that high HbA1c was associated with increasing infarct volume.

Kang-Ho Choi, MD, PhD et al. HbA1c (Glycated Hemoglobin) Levels and Clinical Outcome Post-Mechanical Thrombectomy in Patients With Large Vessel Occlusion. Stroke 2019

Wang, Hong et al. Impact of Elevated Hemoglobin A1c Levels on Functional Outcome in Patients with Acute Ischemic Stroke. Journal of Stroke and Cerebrovascular Diseases, Volume 28, Issue 2, 470 - 476

Sun, Chenghe et al. Glycosylated Hemoglobin A1c Predicts Intracerebral Hemorrhage with Acute Ischemic Stroke Post-Mechanical Thrombectomy. Journal of Stroke and Cerebrovascular Diseases, Volume 29, Issue 9, 105008

Diprose WK, Wang MTM, McFetridge A, et al. Glycated hemoglobin (HbA1c) and outcome following endovascular thrombectomy for ischemic stroke. Journal of NeuroInterventional Surgery 2020;12:30-32.

Clara Hjalmarsson, Karin Manhem, Lena Bokemark, Björn Andersson, "The Role of Prestroke Glycemic Control on Severity and Outcome of Acute Ischemic Stroke", Stroke Research and Treatment, vol. 2014, Article ID 694569, 6 pages, 2014. https://doi.org/10.1155/2014/694569

Response: Thank you for your valuable comments.

It is true that HbA1c is well known predictor for diabetic complications and CV outcomes. Therefore, we tried to emphasize that there are some researches that HbA1c does not predict CV outcomes.

However, I agree that this sentence can confuse the readers.

So, according to the reviewer's advice, it was corrected as follows.

In Introduction, Line 47-50,

Traditionally, HbA1c is recognized as a well-known predictor of diabetic complications and cardiovascular outcomes [4-7]. However, several articles reported that HbA1c was not a good predictor of short-term outcomes, including in-hospital mortality, 6-month major adverse events, and all-cause mortality in the specific population [8,9].

  1. Methods:

The inclusion of TIAs is problematic when comparing the MRS at 3 months since technically this should not contribute to a change in the MRS. Of note, the authors report the baseline severity percentages (non-significant difference), I wonder if the stratification of TIAs would have made a difference since if a major proportion of patients in the < 16 GA group were low NIH of 0-2 and the ones in the high GA group were in the relatively worse baseline like NIH of 5-7 this could have contributed to worse outcomes as well. This stratification applies to NIH of >8 as well considering worse glucose control in days preceding stroke presentation has been associated with worse Ischemic stroke presentations.

Response: Thank you for raising so much important issue.

We fully agree with the reviewer.

So we excluded 184 patients with TIA.

Using a total of 1163 subjects, we conducted all analyses again. (Table 1-6 and Figure 1,2)

Figure 1. Flow diagram of study population. GA, glycated albumin.

Even after excluding TIA patients, GA16% remained a predictor of unfavorable short-term outcomes.

Table 3. Effect of GA on unfavorable short-term outcomes (compared to favorable 3-month mRS 0-2).

Variables

OR

95% CI

P-value

Age, per 1 years

1.034

1.019-1.050

<0.001

Male, sex

0.728

0.525-1.010

0.057

BMI at admission, kg/m2

1.000

0.958-1.043

0.994

Cardiovascular risk factor

 Prior ischemic stroke

1.610

1.137-2.278

0.007

 Hypertension

0.902

0.652-1.248

0.533

 smoking

0.930

0.610-1.418

0.736

 Atrial fibrillation

1.528

0.826-2.826

0.177

Anticoagulation treatment

0.691

0.038-12.562

0.803

Mechanism

LAA

1.674

1.128-2.484

0.011

 SVO

reference

 Cardioembolic

1.215

0.612-2.410

0.578

  Other determined

1.732

0.766-3.921

0.187

  Undetermined

1.454

0.904-2.340

0.123

Laboratory

  Hemoglobin

0.988

0.906-1.078

0.792

  Triglyceride

0.999

0.998-1.001

0.447

  Blood urea nitrogen

1.008

0.993-1.022

0.294

  Albumin

0.854

0.589-1.239

0.405

  hsCRP

1.007

1.001-1.012

0.012

  Glycated albumin <16%

Reference

  Glycated albumin ≥16%

1.374

1.022-1.847

0.035

  Initial Stroke Severity, NIHSS 0-7

reference

                      NIHSS ≥8

1.875

1.345-2.613

<0.001

Note: Adjusted for age, sex, body mass index, prior ischemic stroke, hypertension, smoking, atrial fibrillation, anticoagulation treatment, stroke subtype, hemoglobin, triglyceride, blood urea nitrogen, albumin, hsCRP, initial stroke severity, and glycated albumin.

Abbreviation: BMI, body mass index; hsCRP, high sensitivity C-reactive protein; mRS, modified Rankin Scale; OR, Odds Ratios; CI, confidence intervals; NIHSS, National Institutes of Health Stroke Scale

Results:

  1. Whereas patients with GA≥16% did not have severe initial stroke severity, and shot-term unfavorable functional outcome (3 month mRS, 3-6) was prevalent (25.7% versus 38.6%) (Table 1).

There was no significant difference in the univariate analysis and so this interpretation of not having severe stroke severity cannot be done. Please see above comment in the methods section.

Response: Thank you for your comments.

I fully agree with the reviewer.

According your advices, we excluded patients with TIA and analyzed again.

We remade all the tables.

Table 1. Baseline characteristics according to levels of glycated albumin.

Glycated Albumin<16%

Glycated Albumin≥16%

p-value

No. (%)

645 (55.5)

518 (44.5)

Age, years

67.5±13.5

72.4±12.3

<0.001

Male, sex, %

411 (63.7)

288 (55.6)

0.005

BMI at admission, kg/m2

23.94±3.31

23.85±3.82

0.670

Cardiovascular risk factor

Prior ischemic stroke

109 (16.9)

137 (26.4)

<0.001

Hypertension

364 (56.4)

362 (69.9)

<0.001

Diabetes

92 (14.3)

318 (61.4)

<0.001

Dyslipidemia

81 (12.6)

88 (17.1)

0.031

  Smoking

162 (25.1)

86 (16.6)

<0.001

  Atrial fibrillation

99 (15.3)

101 (19.5)

0.078

Antiplatelet history

148 (22.9)

161 (31.1)

0.002

Anticoagulation history

28 (4.3)

31 (6.0)

0.056

Mechanism

0.001

  LAA

202 (31.6)

209 (40.7)

  SVO

189 (29.5)

101 (19.6)

  Cardioembolic

107 (16.7)

90 (17.5)

  Other determined

32 (5.0)

19 (3.7)

  Undetermined

110 (17.2)

95 (18.5)

Laboratory

  White Blood Cell

7769±2678

7822±2791

0.744

  Hemoglobin, g/dL

13.9±1.9

13.4±2.1

<0.001

  Platelet

233K±68K

224K±72K

0.028

  FBS, mg/dL

117.4±34.6

160.2±73.5

<0.001

  Initial glucose, mg/dL

124.6±35.7

169.4±74.1

<0.001

  HbA1c, %

5.60±0.60

7.00±1.58

<0.001

  HbA1c≥6.5%

37 (5.8)

280 (54.4)

<0.001

  Glycated albumin, %

13.8±1.3

20.1±4.8

<0.001

  GA/HbA1c

2.47±0.33

2.88±0.48

<0.001

  Initial glucose/GA

8.96±2.62

8.28±2.98

<0.001

  Initial glucose/HbA1c

22.14±6.46

23.63±7.88

0.001

  LDL, mg/dL

102.0±32.9

100.7±37.2

0.530

  Total cholesterol, mg/dL

170.1±42.6

164.0±45.2

0.019

  Triglyceride, mg/dL

137.1±105.1

135.1±81.4

0.729

  Blood urea nitrate

16.97±11.05

19.31±9.93

<0.001

  Creatinine

0.94±0.65

1.07±0.89

0.009

  Albumin,

3.98±0.43

3.93±0.47

0.072

  Prothrombin Time

1.08±0.50

1.08±0.22

0.113

  hsCRP

7.90±24.00

14.14±34.32

0.001

Systolic BP, mmHg

151±28

148±28

0.050

Diastolic BP, mmHg

86±15

84±16

0.023

Initial Stroke Severity, NIHSS 0-7

478 (74.1)

404 (78.0)

0.024

                    NIHSS ≥8

167 (25.9)

114 (22.0)

Poor short-term functional outcome,

  (3 month mRS, 3-6)

166 (27.4)

192 (39.7)

<0.001

Note: Abbreviation: BMI, Body mass index; FBS, Fasting blood sugar; GA, Glycated albumin; LDL, Low density lipoprotein; HDL, High density lipoprotein; aPTT, activated prothrombin time; LAA, Large artery atherosclerosis; SVO, Small vessel occlusion; hsCRP, high sensivity C-reactive protein; BP, Blood pressure; IQR, Interquatile ratio; NIHSS, National Institutes of Health Stroke Scale. No. (%) or mean±SD. p Values were calculated by χ2 test for trend in proportion

In the newly conducted analysis, the initial NIHSS was different between the two groups, so the sentence was left as it is.

  1. Results

3.1. Analysis I: Levels of glycated albumin and short-term functional outcome

Among the 1,163 subjects, the mean age was 69.7±13.2 years, and 60.1% were men. The baseline demographic and clinical characteristics are shown in Table 1. A total of 518 patients (44.5%) were included in the GA≥16% group. Patients with GA≥16% were older and had some prevalent conventional vascular risk factors, such as prior ischemic stroke, hypertension, diabetes, and dyslipidemia. However, patients with GA≥16% did not have high initial stroke severity, and unfavorable short-term functional outcomes (3-month mRS, 3-6) were prevalent (27.4% versus 39.7%) (Table 1).

  1. b) After categorizing all patients into the two groups according to short-term functional outcome (favorable, 3-month mRS= 0-2, versus unfavorable, 3-month mRS= 3-6), patients with unfavorable short-term outcome were older, less obese, and more likely to have conventional vascular risk factors including prior ischemic stroke, hypertension, diabetes, and atrial fibrillation (Table 2).

In the final multivariate regression analysis, (Table 3) the authors left out diabetes (which was subgrouped later), use of antiplatelets, anticoagulants, all of which could potentially have influenced the outcomes from a stroke standpoint and were significantly higher in the GA> 16 group based on Table 2. This could pose a challenge in the interpretation of results.

Response: Thank you for your detailed comments.

Generally, variables with p<0.05 in univariate analysis or biological relevance were included in the multivariable model.

However, diabetes was excluded because we thought that GA was multicollinear in diabetes.

However, we fully agree with the reviewer that some variables should be adjusted in the multivariable model.

After excluding 184 patients with TIA, we re-made the Table 2.

In the table 2, variables with p<0.05 in univariate analysis or biological relevance were included in the multivariable model.

Therefore, according to your advices, anticoagulation treatment and albumin were additionally adjusted in multivariate analysis in Table 3.

Table 2. Baseline characteristics according to short-term functional outcomes.

Good functional outcome

3 month mRS, 0-2

Poor functional outcome

3 month mRS, 3-6

p-value

No. (%)

731 (67.1)

358 (32.9)

Age, years

67±13

75±12

<0.001

Male, sex, %

475 (65.0)

176 (49.2)

<0.001

BMI at admission, kg/m2

24.09±3.38

23.53±3.88

0.015

BMI Quartile, Q 1

166 (23.0)

101 (28.6)

0.015*

Q 2

170 (23.6)

91 (25.8)

            Q 3

195 (27.0)

83 (23.5)

Q 4

190 (26.4)

78 (22.1)

Cardiovascular risk factor

  Prior ischemic stroke

128 (17.5)

101 (28.2)

<0.001

 Hypertension

429 (58.7)

240 (67.0)

0.008

 Diabetes

244 (33.4)

138 (38.5)

0.093

 Dyslipidemia

89 (12.2)

59 (16.5)

0.051

  Smoking

176 (24.1)

47 (13.1)

<0.001

  Atrial fibrillation

87 (11.9)

94 (26.3)

<0.001

Antiplatelet history

178 (24.4)

111 (31.0)

0.055

Anticoagulation history

18 (2.5)

33 (9.2)

<0.001

Mechanism

<0.001

  LAA

250 (34.5)

135 (37.8)

  SVO

223 (30.8)

57 (16.0)

  Cardioembolic

100 (13.8)

82 (23.0)

  Other determined

32 (4.4)

15 (4.2)

  Undetermined

119 (16.4)

68 (19.0)

Laboratory

  White Blood Cell

7676±2542

8019±3117

0.071

  Hemoglobin, g/dL

13.9±1.8

13.2±2.2

<0.001

  Platelet

230K±64K

228K±80K

0.570

  FBS, mg/dL

134.4±59.6

139.6±59.1

0.174

  Initial glucose, mg/dL

143.8±61.1

145.6±59.9

0.656

  HbA1c, %

6.22±1.33

6.25±1.37

0.678

  HbA1c≥6.5%

202 (27.7)

102(28.7)

0.715

  Glycated albumin, %

16.30±4.58

17.20±4.61

0.002

  Glycated albumin ≥16%

292 (39.9)

192 (53.6)

<0.001

  GA/HbA1c

2.62±0.40

2.74±0.51

<0.001

  Initial glucose/GA

8.85±2.47

8.56±2.77

0.073

  Initial glucose/HbA1c

22.89±6.42

23.26±7.08

0.383

  LDL, mg/dL

102.5±35.1

99.9±35.5

0.257

  Total cholesterol, mg/dL

168.8±43.0

164.3±45.5

0.118

  Triglyceride, mg/dL

144.7±96.7

121.8±96.5

<0.001

  Blood urea nitrate

17.15±9.52

19.68±11.18

<0.001

  Creatinine

0.98±0.73

1.07±0.88

0.129

  Albumin,

4.00±0.40

3.85±0.52

<0.001

  Prothrombin Time

1.08±0.49

1.09±0.19

0.714

  hsCRP

7.63±23.63

17.10±38.31

<0.001

Systolic BP, mmHg

149±27

150±29

0.686

Diastolic BP, mmHg

85±15

84±17

0.095

Initial Stroke Severity, NIHSS 0-7

592 (81.0)

237 (66.2)

<0.001

     NIHSS ≥8

139 (19.0)

121 (33.8)

Note: Abbreviation: BMI, Body mass index; FBS, Fasting blood sugar; GA, Glycated albumin; LDL, Low density lipoprotein; HDL, High density lipoprotein; aPTT, activated prothrombin time; LAA, Large artery atherosclerosis; SVO, Small vessel occlusion; hsCRP, high sensivity C-reactive protein; BP, Blood pressure; IQR, Interquatile ratio; NIHSS, National Institutes of Health Stroke Scale. No. (%) or mean±SD. p Values were calculated by χ2 test for trend in proportion. *Linear by linear association for trend.

Table 3. Effect of GA on unfavorable short-term outcomes (compared to favorable 3-month mRS 0-2).

Variables

OR

95% CI

P-value

Age, per 1 years

1.034

1.019-1.050

<0.001

Male, sex

0.728

0.525-1.010

0.057

BMI at admission, kg/m2

1.000

0.958-1.043

0.994

Cardiovascular risk factor

 Prior ischemic stroke

1.610

1.137-2.278

0.007

 Hypertension

0.902

0.652-1.248

0.533

 smoking

0.930

0.610-1.418

0.736

 Atrial fibrillation

1.528

0.826-2.826

0.177

Anticoagulation treatment

0.691

0.038-12.562

0.803

Mechanism

LAA

1.674

1.128-2.484

0.011

 SVO

reference

 Cardioembolic

1.215

0.612-2.410

0.578

  Other determined

1.732

0.766-3.921

0.187

  Undetermined

1.454

0.904-2.340

0.123

Laboratory

  Hemoglobin

0.988

0.906-1.078

0.792

  Triglyceride

0.999

0.998-1.001

0.447

  Blood urea nitrogen

1.008

0.993-1.022

0.294

  Albumin

0.854

0.589-1.239

0.405

  hsCRP

1.007

1.001-1.012

0.012

  Glycated albumin <16%

Reference

  Glycated albumin ≥16%

1.374

1.022-1.847

0.035

  Initial Stroke Severity, NIHSS 0-7

reference

                      NIHSS ≥8

1.875

1.345-2.613

<0.001

Note: Adjusted for age, sex, body mass index, prior ischemic stroke, hypertension, smoking, atrial fibrillation, anticoagulation treatment, stroke subtype, hemoglobin, triglyceride, blood urea nitrogen, albumin, hsCRP, initial stroke severity, and glycated albumin.

Abbreviation: BMI, body mass index; hsCRP, high sensitivity C-reactive protein; mRS, modified Rankin Scale; OR, Odds Ratios; CI, confidence intervals; NIHSS, National Institutes of Health Stroke Scale

  1. Table 3: Can the authors explain how these specific variables were chosen?

Response: Thank you for your comments.

Generally, variables with p<0.05 in univariate analysis or biological relevance were included in the multivariable model.

However, diabetes was excluded because we thought that GA was multicollinear in diabetes.

We fully agree with the reviewer that anticoagulation treatment should be adjusted.

After excluding 184 patients with TIA, we re-made Table 2.

Variables with p<0.05 in univariate analysis or biological relevance were included in the multivariable model.

Therefore, according to your advices, anticoagulation treatment and albumin were additionally adjusted in multivariate analysis in Table 3.

Table 3. Effect of GA on unfavorable short-term outcomes (compared to favorable 3-month mRS 0-2).

Variables

OR

95% CI

P-value

Age, per 1 years

1.034

1.019-1.050

<0.001

Male, sex

0.728

0.525-1.010

0.057

BMI at admission, kg/m2

1.000

0.958-1.043

0.994

Cardiovascular risk factor

 Prior ischemic stroke

1.610

1.137-2.278

0.007

 Hypertension

0.902

0.652-1.248

0.533

 smoking

0.930

0.610-1.418

0.736

 Atrial fibrillation

1.528

0.826-2.826

0.177

Anticoagulation treatment

0.691

0.038-12.562

0.803

Mechanism

LAA

1.674

1.128-2.484

0.011

 SVO

reference

 Cardioembolic

1.215

0.612-2.410

0.578

  Other determined

1.732

0.766-3.921

0.187

  Undetermined

1.454

0.904-2.340

0.123

Laboratory

  Hemoglobin

0.988

0.906-1.078

0.792

  Triglyceride

0.999

0.998-1.001

0.447

  Blood urea nitrogen

1.008

0.993-1.022

0.294

  Albumin

0.854

0.589-1.239

0.405

  hsCRP

1.007

1.001-1.012

0.012

  Glycated albumin <16%

Reference

  Glycated albumin ≥16%

1.374

1.022-1.847

0.035

  Initial Stroke Severity, NIHSS 0-7

reference

                      NIHSS ≥8

1.875

1.345-2.613

<0.001

Note: Adjusted for age, sex, body mass index, prior ischemic stroke, hypertension, smoking, atrial fibrillation, anticoagulation treatment, stroke subtype, hemoglobin, triglyceride, blood urea nitrogen, albumin, hsCRP, initial stroke severity, and glycated albumin.

Abbreviation: BMI, body mass index; hsCRP, high sensitivity C-reactive protein; mRS, modified Rankin Scale; OR, Odds Ratios; CI, confidence intervals; NIHSS, National Institutes of Health Stroke Scale

  1. The information on GA and BMI as stated prior is irrelevant in this manuscript:

He X, Mo Y, Ma X, Ying L, Zhu W, Wang Y, Bao Y, Zhou J. Associations of body mass index with glycated albumin and glycated albumin/glycated hemoglobin A1c ratio in Chinese diabetic and non-diabetic populations. Clin Chim Acta. 2018 Sep;484:117-121. doi: 10.1016/j.cca.2018.05.044. Epub 2018 May 24. PMID: 29802831.

Miyashita Y, Nishimura R, Morimoto A, Matsudaira T, Sano H, et al. (2007) Glycated albumin is low in obese, type 2 diabetic patients. Diabetes Res Clin Pract 78: 51–55.

Nishimura R, Kanda A, Sano H, Matsudaira T, Miyashita Y, et al. (2006) Glycated albumin is low in obese, non-diabetic children. Diabetes Res Clin Pract 71: 334–338.

Koga M, Matsumoto S, Saito H, Kasayama S (2006) Body mass index negatively influences glycated albumin, but not glycated hemoglobin, in diabetic patients. Endocr J 53: 387–391.

Response: Thank you for your valuable comments.

I know there were already several articles showing the inverse relation of GA with BMI.

However, few articles have dealt with the relationship between BMI and GA in patients with Acute ischemic stroke.

The authors thought this was very interesting and wanted to emphasize it.

However, as the reviewer points out, the authors fully agree that this title does not serve the purpose of this paper. Thank you for raising so much important comments.

So we changed the title as follows.

Glycated Albumin, a Novel Biomarker for Short-Term Stroke Outcome and Its Inversed Association with Body Mass Index

àGlycated Albumin, a Novel Biomarker for Short-Term Functional Outcomes In Acute Ischemic Stroke

In addition, we moved “Table 5. Correlations between body mass index and Glycemic control indicators” to the Supplementary table.

Discussion:

  1. (1) patients with a higher GA level 202 (GA≥16%) had unfavorable short-term functional outcomes at 3 month after stroke onset; 203 (2) a higher GA and higher GA/A1c ratio was significantly associated with unfavorable 204 short-term functional outcome at 3 month after stroke onset, but HbA1c was not; and (3) a higher GA level was associated with unfavorable short-term functional outcome at 3 months after stroke onset only in patients without diabetes.

Based on the aforementioned comments in previous sections, the inference 1 cannot be sustained, inference 3 is a result of subgroup testing without multiple correction. The authors can consider the use of Bonferroni correction or similar strategies to overcome this issue.

Response: Thank you for your comments.

According your advices, we excluded patients with TIA and analyzed again.

We remade all the tables and figures.

In Table 1 and Table 2, since dependent variable is categorical and binary, we use chi-square test or Student’s t-test.

In supplementary Table, since sample size is large enough, we used ANOVA and ran Scheffe test to find out which pairs of means are significant. The Scheffe test is a post-hoc analysis used in Analysis of Variance. a,b)The same letters indicated non-significant difference between groups based on Scheffe multiple comparison test.

Supplementary Table. Correlations between body mass index and Glycemic control indicators.

BMI, Q1

BMI, Q2

BMI, Q3

BMI, Q4

p-value

No. (%)

331 (24.6)

328 (24.4)

339 (25.2)

333 (24.7)

Age, years

72±14a

71±13 a

68±12 b

67±13 b

<0.0011)

Diabetes

85 (29.3)

94 (33.7)

109 (36.9)

118 (41.4)

0.002*

Laboratory

FBS, mg/dL

130.3±53.6a

139.4±66.7 a

132.4±50.5 a

140.6±58.1 a

0.1091)

Initial glucose, mg/dL

139.1±56.0 a

145.5±65.3 a

143.2±56.3 a

150.5±62.0 a

0.1851)

HbA1c, g/dL

5.99±1.19 a

6.25±1.39 a,b

6.25±1.34 a.b

6.35±1.36 b

0.0041)

 HbA1c≥6.5%

61 (21.4)

72 (25.9)

82 (27.9)

100 (35.2)

<0.001*

 Glycoalbumin, Quartiles

0.003*

  Glycoalbumin, 1Q

47 (16.3)

56 (20.1)

90 (30.5)

78 (27.4)

  Glycoalbumin, 2Q

72 (25.0)

79 (28.3)

64 (21.7)

76 (26.7)

  Glycoalbumin, 3Q

93 (32.3)

74 (26.5)

70 (23.7)

57 (20.0)

  Glycoalbumin, 4Q

76 (26.4)

70 (25.1)

71 (24.1)

74 (26.0)

 GA/HbA1c

2.77±0.51 a

2.69±0.48 b

2.58±0.39 b

2.57±0.41 b

<0.0011)

Abbreviation: BMI, Body mass index; FBS, Fasting blood sugar; IQR, Interquatile ratio. No. (%) or mean±SD. p Values were calculated by χ2 test for trend in proportion. *Linear by linear association for trend. 1)Statistical significances were tested by Oneway analysis of variances among groups. a,b)The same letters indicated non-significant difference between groups based on Scheffe multiple comparison test.

  1. Similar to our study, some previous articles reported that HbA1c was not an independent predictor of short-term outcomes [4,5]. In a retrospective study of 317 diabetic patients with acute coronary syndrome, HbA1c levels before admission are not related to short-term cardiovascular outcome including in-hospital mortality, 6-month major adverse cardiovascular events (MACE), and all-cause mortality [4]. In an observational multicenter study of 608 patients with acute myocardial infarction, HbA1c was not associated with 7 day mortality and 30-day mortality [5]. However, the results are still controversial. Some authors demonstrated that HbA1c was a potential indicator for in-hospital death in patients with acute coronary syndrome [12] and was a good predictor of acute and long-term mortality in patients with AIS [13]. Furthermore, in a total of 534 subjects with AIS treated with mechanical thrombectomy, HbA1c≥6.5% was an independent predictor of a poor outcome at 3 months after AIS [14].

As stated prior including references this is not accurate from a stroke standpoint. Can the authors respond why cardiovascular outcomes were chosen instead of stroke outcomes when there is good data on stroke-based outcomes and A1c relation? Please see information from the introduction and the references associated with stroke outcomes and A1c. Of note, the one reference from an AIS standpoint actually showed A1cs relation with poor outcome. Reference 14. This is one of the fewer studies to show A1c to not be related to poor outcomes, but the interpretation is limited due to the aforementioned limitations and potential confounding.

Response: Thank you for your valuable comments.

It is true that HbA1c is well known predictor for diabetic complications and CV outcomes. Therefore, we tried to emphasize that there are some researches that HbA1c does not predict CV outcomes.

However, I agree that this sentence can confuse the readers.

So, according to the reviewer's advice, it was corrected as follows.

In Introduction, Line 47-50,

Traditionally, HbA1c is recognized as a well-known predictor of diabetic complications and cardiovascular outcomes [4-7]. However, several articles reported that HbA1c was not a good predictor of short-term outcomes, including in-hospital mortality, 6-month major adverse events, and all-cause mortality in the specific population [8,9].

According to your comments, we thought that we have to tone down our suggestion.

Therefore, we changed the sentences as follows.

In Discussion,

Interestingly, we found that when GA was replaced by HbA1c (either as a continuous variable or as a binary variable with a cutoff of HbA1c ≥6.5%) in the same model, HbA1c failed to prove the significant association in this study. We should not generalize this result. Conventionally, HbA1c has been a well-recognized predictor of cardiovascular outcomes in previous reports. Some authors demonstrated that HbA1c was a potential indicator for in-hospital death in patients with acute coronary syndrome [16] and was a good predictor of acute and long-term mortality in patients with AIS [17]. Furthermore, in a total of 534 subjects with AIS treated with mechanical thrombectomy, HbA1c≥6.5% was an independent predictor of a poor outcome at 3 months after AIS [5]. However, the results are still controversial. Similar to our study, some prior articles reported that HbA1c was not an independent predictor of short-term outcomes [8,9]. In a retrospective study of 317 diabetic patients with acute coronary syndrome, HbA1c levels before admission were not related to short-term cardiovascular outcomes, including in-hospital mortality, 6-month major adverse cardiovascular events (MACE), and all-cause mortality [8]. In an observational multicenter study of 608 patients with acute myocardial infarction, HbA1c was not associated with 7-day mortality or 30-day mortality [9]. We cannot explain the exact pathomechanism, and we suggest that GA might also be an important indicator along with HbA1c [18].

  1. In addition, the effect of BMI on GA might also be considered to evaluate the role of 259 GA on stroke outcome. According to previous reports, in spite of the inconsistent results, 260 obesity seems to be negatively associated with GA or GA/A1c [17-20]. In this study, the 261 BMI showed a significantly positive correlation with HbA1c, while BMI negatively corre-262 lated with GA or GA/A1c (supplementary table). At present, the reasons for the negative 263 influence of BMI on GA are not clear. However, possible explanation is as follows: (1) 264 Salas-Salvado et al. demonstrated that albumin levels are lower in obese subjects than 265 non-obese counterparts [21]. But Miyashita et al. found that obese children had higher 266 albumin concentration than non-obese children [22], and Koga et al. reported no correla-267 tion between BMI and albumin levels [18]. (2) turnover of albumin may be increased in 268 obese patients. Since chronic low-grade systemic inflammation is involved in obese sub-269 jects, inflammation might increase the catabolic rate of albumin and decrease the rate of 270 albumin synthesis. The authors did not provide the exact mechanism, but hypothesized 271 that inflammation represented by elevated hs-CRP was significantly associated with BMI, 272 which could result in lower GA [18]. In our study, when BMI is increasing, while GA 273 showed a decreasing trend, albumin showed an increasing trend. Therefore, we suggest 274 that it affects albumin turnover rather than the albumin concentration itself. Although we 275 have not been able to draw any conclusions through this paper, I think it suggested an 276 important key finding for the next study.

As stated, this discussion of BMI and GA relation does not seem relevant to their primary hypothesis and goal of this paper.

Response: Thank you for raising so much important issue.

We fully agree with the reviewer.

Therefore, we changed the title as follows.

Glycated Albumin, a Novel Biomarker for Short-Term Stroke Outcome and Its Inversed Association with Body Mass Index

àGlycated Albumin, a Novel Biomarker for Short-Term Functional Outcomes In Acute Ischemic Stroke

In addition, we moved “Table 5. Correlations between body mass index and Glycemic control indicators” to the Supplementary table.

  1. A major strength is that this is the first study to evaluate the effect of GA on short- term functional outcome in patients with AIS.

The below study was published on GA and outcomes in AIS late last year and so this sentence can bebe tempered.

Lee SH, Kim Y, Park SY, Kim C, Kim YJ, Sohn JH. Pre-Stroke Glycemic Variability Estimated by Glycated Albumin Is Associated with Early Neurological Deterioration and Poor Functional Outcome in Prediabetic Patients with Acute Ischemic Stroke. Cerebrovasc Dis. 2021;50(1):26-33. doi: 10.1159/000511938. Epub 2020 Dec 1. PMID: 33260185.

Response: Thank you for your comments.

We changed the sentence as follows.

A major strength is that this is the first study to evaluate the effect of GA on short- term functional outcome in patients with AIS.

  • The major strength is that this is a multicenter study with a relatively large sample size.
  1. Grammatical / editing / Minor Comments

Response: Thank you for your comments.

In fact, this manuscript had been edited by a professional English proofreading company.

But I had it revised by a professional English proofreading company again. I attached the “Edit certification” from AJE.

We thank the reviewer for the valuable feedback, which enabled us to greatly improve the quality of our manuscript.

“growing interest to use a new biomarker such as Glycated albumin (GA), but data are limited in acute ischemic stroke”

“We explored the impact of GA for short-term 18 functional outcome using the modified Rankin Scale (mRS) at 3 months compared to glycated hemoglobin (HbA1c);”

Response: Thank you for your comments.

We changed the sentence as follows.

There is growing interest in the use of new biomarkers such as glycated albumin (GA), but data are limited in acute ischemic stroke. We explored the impact of GA on short-term functional outcomes as measured using the modified Rankin Scale (mRS) at 3 months compared to glycated hemoglobin (HbA1c).

“The 595 patients (44.2%) were included in the GA≥16% group.”

Response: Thank you for your comments.

We changed the sentence as follows.

Results: A total of 518 patients (44.5%) were included in the GA≥16% group.

“GA level might be a novel prognostic biomarker than HbA1c for short-term stroke outcome. Although the impact of GA is undervalued in the current stroke guidelines, monitoring GA should be considered in addition to HbA1c;”

Response: Thank you for your comments.

We changed the sentence as follows.

Conclusions: In light of these results, GA level might be a novel prognostic biomarker than HbA1c for short-term stroke outcome. Although the impact of GA is undervalued in the current stroke guidelines, GA monitoring should be considered in addition to HbA1c monitoring.

“In this regard, there have been some needs for detecting a novel index for diagnosing and managing glycemic control in covering the short-term effect in many critical illnesses.”

Response: Thank you for your comments.

We changed the sentence as follows.

In this regard, there has been some need to detect a novel index for diagnosing and managing glycemic control in covering the short-term effect in many critical illnesses.

“fifty-eight subjects who were not evaluate for GA levels were excluded”         

Response: Thank you for your comments.

We changed the sentence as follows.

Among a total of 1,405 patients (541 patients from Kangdong Sacred Heart Hospital and 864 patients from Chuncheon Sacred Heart Hospital), fifty-eight subjects who were not evaluated for GA levels and 184 patients with TIA were excluded.

Further well-designed study to investigate the relationship between glycemic control indicators and glucose tolerance status should be needed.

Response: Thank you for your comments.

We changed the sentence as follows.

Further well-designed studies to investigate the relationship between glycemic control indicators and glucose tolerance status are needed.

Patients with GA≥16% were older and had some prevalent conventional vascular risk factors, such as prior ischemic stroke, hypertension, diabetes, smoking history, and atrial fibrillation. Whereas patients with GA≥16% did not have severe initial stroke severity, and shot-term unfavorable functional outcome (3 month mRS, 3-6) was prevalent (25.7% versus 38.6%)

Response: Thank you for your valuable comments.

We changed the sentence as follows.

Patients with GA≥16% were older and had some prevalent conventional vascular risk factors, such as prior ischemic stroke, hypertension, diabetes, and dyslipidemia. However, patients with GA≥16% did not have high initial stroke severity, and unfavorable short-term functional outcomes (3-month mRS, 3-6) were prevalent (27.4% versus 39.7%) (Table 1).

  1. Overall:

Multiple issues with Grammar - article usage, punctuations and sentence structuring through the manuscript.  Only a few examples have been elicited above. Before re-submitting this manuscript, the authors could consider having it proofread by someone well-versed in written English or possibly utilize an editing resource. Brevity is also an issue with this manuscript, with an extensive amount of irrelevant information including some on BMI and GA which doesn’t pertain to the primary aim of this manuscript. Some of the biggest limitations pertain to the methodological aspects and the confounding as discussed above and the inaccurate discussion regarding A1c and outcomes.

Response: Thank you for your comments.

We revised point-by-point as your suggestions. 

We thank the reviewer for the valuable feedback, which enabled us to greatly improve the quality of our manuscript.

Reviewer 2 Report

The present manuscript contains the results of a study aimed at confirming the usefulness of glycated albumin as a new biomarker for short-term stroke outcome. The authors showed a statistically significant difference in the amount of glycated albumin in both groups. Limitations of the study were properly stated.  The authors present interesting clinical data here, which they discussed relatively well. The findings presented certainly need to be verified by another study with a larger number of patients. Nevertheless, I believe that these results are interesting and may serve as a basis for further research. As I have already mentioned, the results are, in my opinion, properly described and linked to the results of other relevant studies.

Minor point: There is an asterisk in Table 2 that is not explained anywhere.  

Author Response

Journal : Brain Sciences (ISSN 2076-3425)

Manuscript ID : brainsci-1108388

Type : Article

Title : Glycated Albumin, a Novel Biomarker for Short-Term Stroke Outcome and Its Inversed Association with Body Mass Index

We thank the reviewer for the valuable feedback, which enabled us to greatly improve the quality of our manuscript. Here we address the reviewer’ concerns point-by-point and explain, where applicable, how we have been able to incorporate the suggestions into our revision. The reviewer’s original comments are copied below in italics.

The present manuscript contains the results of a study aimed at confirming the usefulness of glycated albumin as a new biomarker for short-term stroke outcome. The authors showed a statistically significant difference in the amount of glycated albumin in both groups. Limitations of the study were properly stated.  The authors present interesting clinical data here, which they discussed relatively well. The findings presented certainly need to be verified by another study with a larger number of patients. Nevertheless, I believe that these results are interesting and may serve as a basis for further research. As I have already mentioned, the results are, in my opinion, properly described and linked to the results of other relevant studies.

Minor point: There is an asterisk in Table 2 that is not explained anywhere.  

Response: Thank you for your valuable comments.

We added the sentences as follows.

Table 2. Baseline characteristics according to short-term functional outcomes.

Good functional outcome

3 month mRS, 0-2

Poor functional outcome

3 month mRS, 3-6

p-value

No. (%)

731 (67.1)

358 (32.9)

Age, years

67±13

75±12

<0.001

Male, sex, %

475 (65.0)

176 (49.2)

<0.001

BMI at admission, kg/m2

24.09±3.38

23.53±3.88

0.015

BMI Quartile, Q 1

166 (23.0)

101 (28.6)

0.015*

Q 2

170 (23.6)

91 (25.8)

            Q 3

195 (27.0)

83 (23.5)

Q 4

190 (26.4)

78 (22.1)

Cardiovascular risk factor

  Prior ischemic stroke

128 (17.5)

101 (28.2)

<0.001

 Hypertension

429 (58.7)

240 (67.0)

0.008

 Diabetes

244 (33.4)

138 (38.5)

0.093

 Dyslipidemia

89 (12.2)

59 (16.5)

0.051

  Smoking

176 (24.1)

47 (13.1)

<0.001

  Atrial fibrillation

87 (11.9)

94 (26.3)

<0.001

Antiplatelet history

178 (24.4)

111 (31.0)

0.055

Anticoagulation history

18 (2.5)

33 (9.2)

<0.001

Mechanism

<0.001

  LAA

250 (34.5)

135 (37.8)

  SVO

223 (30.8)

57 (16.0)

  Cardioembolic

100 (13.8)

82 (23.0)

  Other determined

32 (4.4)

15 (4.2)

  Undetermined

119 (16.4)

68 (19.0)

Laboratory

  White Blood Cell

7676±2542

8019±3117

0.071

  Hemoglobin, g/dL

13.9±1.8

13.2±2.2

<0.001

  Platelet

230K±64K

228K±80K

0.570

  FBS, mg/dL

134.4±59.6

139.6±59.1

0.174

  Initial glucose, mg/dL

143.8±61.1

145.6±59.9

0.656

  HbA1c, %

6.22±1.33

6.25±1.37

0.678

  HbA1c≥6.5%

202 (27.7)

102(28.7)

0.715

  Glycated albumin, %

16.30±4.58

17.20±4.61

0.002

  Glycated albumin ≥16%

292 (39.9)

192 (53.6)

<0.001

  GA/HbA1c

2.62±0.40

2.74±0.51

<0.001

  Initial glucose/GA

8.85±2.47

8.56±2.77

0.073

  Initial glucose/HbA1c

22.89±6.42

23.26±7.08

0.383

  LDL, mg/dL

102.5±35.1

99.9±35.5

0.257

  Total cholesterol, mg/dL

168.8±43.0

164.3±45.5

0.118

  Triglyceride, mg/dL

144.7±96.7

121.8±96.5

<0.001

  Blood urea nitrate

17.15±9.52

19.68±11.18

<0.001

  Creatinine

0.98±0.73

1.07±0.88

0.129

  Albumin,

4.00±0.40

3.85±0.52

<0.001

  Prothrombin Time

1.08±0.49

1.09±0.19

0.714

  hsCRP

7.63±23.63

17.10±38.31

<0.001

Systolic BP, mmHg

149±27

150±29

0.686

Diastolic BP, mmHg

85±15

84±17

0.095

Initial Stroke Severity, NIHSS 0-7

592 (81.0)

237 (66.2)

<0.001

     NIHSS ≥8

139 (19.0)

121 (33.8)

Note: Abbreviation: BMI, Body mass index; FBS, Fasting blood sugar; GA, Glycated albumin; LDL, Low density lipoprotein; HDL, High density lipoprotein; aPTT, activated prothrombin time; LAA, Large artery atherosclerosis; SVO, Small vessel occlusion; hsCRP, high sensivity C-reactive protein; BP, Blood pressure; IQR, Interquatile ratio; NIHSS, National Institutes of Health Stroke Scale. No. (%) or mean±SD. p Values were calculated by χ2 test for trend in proportion. *Linear by linear association for trend was used.

Round 2

Reviewer 1 Report

Thank you for incorporating the suggested changes.